# Biphasic Force-Regulated Phosphorylation Site Exposure and Unligation of ERM Bound with PSGL-1: A Novel Insight into PSGL-1 Signaling via Steered Molecular Dynamics Simulations

**DOI:** 10.3390/ijms21197064

**Published:** 2020-09-25

**Authors:** Jingjing Feng, Yan Zhang, Quhuan Li, Ying Fang, Jianhua Wu

**Affiliations:** Institute of Biomechanics/School of Bioscience and Bioengineering, South China University of Technology, Guangzhou 510006, China; 201720143514@mail.scut.edu.cn (J.F.); 201721044247@mail.scut.edu.cn (Y.Z.); liqh@scut.edu.cn (Q.L.)

**Keywords:** PSGL-1, ERM protein, Syk, ITAM-like motif, leukocyte, molecular dynamics simulations

## Abstract

The PSGL-1-actin cytoskeleton linker proteins ezrin/radixin/moesin (ERM), an adaptor between P-selectin glycoprotein ligand-1 (PSGL-1) and spleen tyrosine kinase (Syk), is a key player in PSGL-1 signal, which mediates the adhesion and recruitment of leukocytes to the activated endothelial cells in flow. Binding of PSGL-1 to ERM initials intracellular signaling through inducing phosphorylation of Syk, but effects of tensile force on unligation and phosphorylation site exposure of ERM bound with PSGL-1 remains unclear. To answer this question, we performed a series of so-called “ramp-clamp” steered molecular dynamics (SMD) simulations on the radixin protein FERM domain of ERM bound with intracellular juxtamembrane PSGL-1 peptide. The results showed that, the rupture force of complex pulled with constant velocity was over 250 pN, which prevented the complex from breaking in front of pull-induced exposure of phosphorylation site on immunoreceptor tyrosine activation motif (ITAM)-like motif of ERM; the stretched complex structure under constant tensile forces <100 pN maintained on a stable quasi-equilibrium state, showing a high mechano-stabilization of the clamped complex; and, in consistent with the force-induced allostery at clamped stage, increasing tensile force (<50 pN) would decrease the complex dissociation probability but facilitate the phosphorylation site exposure, suggesting a force-enhanced biophysical connectivity of PSGL-1 signaling. These force-enhanced characters in both phosphorylation and unligation of ERM bound with PSGL-1 should be mediated by a catch-slip bond transition mechanism, in which four residue interactions on binding site were involved. This study might provide a novel insight into the transmembrane PSGL-1 signal, its biophysical connectivity and molecular structural basis for cellular immune responses in mechano-microenvironment, and showed a rational SMD-based computer strategy for predicting structure-function relation of protein under loads.

## 1. Introduction

As a critical actor for recruitment of flowing leukocytes to vascular injury sites, P-selectin glycoprotein ligand-1 (PSGL-1), a transmembrane protein on leukocytes, mediates tethering and rolling of flowing leukocytes through binding to L-, P- or E-selectins, the three selectin family members on the endothelial cells [1,2,3], triggers a series of intracellular signal transductions, including recruitment and phosphorylation of Src kinases and spleen tyrosine kinase (Syk) [4,5]. The PSGL-1 signal induces extension and activation of αLβ2 integrin and further makes rolling of the circulating leukocyte slow, leading to leukocyte recruitment and activation [6,7,8,9,10].

In PSGL-1 signaling, Syk is activated normally by the immunoreceptor tyrosine activation motif (ITAM, Yxx(I/L)x(6–12)Yxx(I/L)), or through its dual Src homology 2 (SH2) domain bound with the tyrosine phosphorylated ITAM [7,11,12,13]. The phosphorylation of the ITAM-like motif on the ezrin/radixin/moesin (ERM) proteins is critical for recruiting Syk molecules, activating Src family kinases, triggering cellular calcium responses and thereby promoting leukocyte proliferation, migration and invasion [4]. As a key player in downstream signaling, the ITAM sequence has been found in intracellular regions of other receptors or transmembrane proteins, such as T and B cell receptor as well as Fc receptor [14,15]. It has been demonstrated that the exposure and phosphorylation of the ITAM sequence phosphorylation site are crucial for Syk activation, some non-traditional ITAM sequences (ITAM-like motif) also have functions similar to traditional ITAM sequences [16,17], and a single tyrosine residue phosphorylation may be sufficient for recruitment of Syk [18,19]. Additionally, studies have shown that the slow rolling of leukocytes in flow requires the PSGL-1 cytoplasmic domain [8,13,20], which mediates Syk activation via binding to the ITAM-like motif of the ezrin/radixin/moesin (ERM) proteins family [3,4,21]. Functions of ERM protein are associated with cell surface protrusions such as microvilli, filopodia, microspikes, retraction fibers and membrane ruffling [22]. However, it has been demonstrated that ERM regulates PSGL-1 signaling by mediating the connection between the actin cytoskeleton and the plasma membrane, and the N-terminal FERM domain of ERM can bind to not only PSGL-1 but also other adhesion molecules such as Na+/H+ exchanger regulatory factor (NHERF) and Rho GDP-dissociation inhibitor [23].

Recent studies have revealed how the radixin FERM domain binds target proteins at the atomic level. The solved crystal structure for complex of mouse source PSGL-1 cytoplasmic tail and radixin protein FERM domain (residues 1–313) (Figure 1) [24] has showed that, in the complex, the juxtamembrane region of the PSGL-1 cytoplasmic tail consists of 16 amino acids (residues 402–417), which forms a stable conformation bound with the FERM domain, which consists of three subdomains (A, B and C); in FERM domain, the C subdomain bound with the PSGL-1 peptide is folded into a seven-stranded β-sandwich enclosing a hydrophobic core with the C-terminal α-helix packed between strands β5C and β1C; the PSGL-1 β-strand forms an antiparallel β-β association with strand β5C from the C subdomain, and twelve residues are involved in hydrogen bonding across interface between the PSGL-1 peptide and the FERM C subdomain protein (Figure 1A); the ITAM-like motif (residues 191–208) acts as a hinge region to link the subdomain B and C, and the two phosphorylation sites Y191 and Y205 are buried by subdomain B and C, respectively (Figure 1B). Such embedding situation of the phosphorylation sites Y191 and Y205 suggests that a steric hindrance effect exists in both the exposure of the phosphorylation sites on ITAM-like motif and the recruitment of Syk, in preventing leukocyte from over-activation through the PSGL-1 signal pathway under hemodynamics environment.

Increasing evidences, such as the “catch-slip bond” transition in mediating selectin-PSGL-1 interaction-induced rolling of leukocyte and the force-facilitated calcium bursting of neutrophils on immobile P-selection as well as the force-regulated β2-integrin activation of leukocyte [25,26,27,28,29], show a mechano-chemical PSGL-1 signaling in recruitment of leukocyte to injured vessel sites in flow, but regulation of fluid shear stress on intracellular PSGL-1 signaling and its biophysical connectivity remains unclear. However, it is rational that, in flow, a stable mechano-chemical PSGL-1 signaling may require a catch bond mechanism for binding of PSGL-1 to FERM, and a better complex strength could make a force-induced exposure of the phosphorylation sites in FERM bound with PSGL-1 peptide possible. On the other hand, as a preferred tool in studying receptor-ligand interaction at atom level, the molecular dynamics (MD) simulation can provide various kinetic structural information for either free- or ligated-protein [24,30,31]. Thus, with the crystal structure of PSGL-1 peptide-bound FERM protein (Figure 1) [24], we here performed steered molecular dynamics (SMD) simulation with “ramp-clamp” mode to examine the force-regulated dissociation of PSGL-1 from FERM in various mechano-microenvironments (Appendix A). Our results demonstrated a possible mechano-regulation mechanism and its structural basis for interaction of PSGL-1 with FERM as well as a force-induced exposure of the phosphorylation sites in FERM bound with PSGL-1 peptide. The present results provide a novel insight into PSGL-1 signal and its biophysical connectivity in leukocyte immune responses under mechano- microenvironments.

## 2. Results

### 2.1. The Stable and Rational Conformation of PSGL-1/ERM Complex at Equilibrium

We herein obtained three equilibrated structures of PSGL-1/ERM complex by performing a system equilibrium of 100 ns thrice (Figure 1), along a same protocol of energy minimization and hypothesis that the complex was equilibrated if the time courses of the temperature, total energy and root mean square deviation (RMSD) of heavy atoms were fluctuated on their respective stable levels with small relative derivations (Figure 2A). The three complexes in equilibrium were denoted by the 1st, 2nd and 3rd equilibrated complexes, which came from the 1st, 2nd and 3rd runs, respectively. Being across PSGL-1/FERM interface, H-bonds with mean number of 8.3, 6.5 and 8.3 were contributed to the three equilibrated complexes with interaction energies of −269, −227 and −225 kcal mol^−1^, respectively (Figure 2B). It showed that the 1st equilibrated structures should be more stable for their lower interaction energies and stronger hydrogen bonding across PSGL-1/FERM interface, in comparison with the 2nd and 3rd equilibrated structures, suggesting the two affinity states of PSGL-1 binding to FERM, and possibly coming from the high flexibility of the β sheet-type structure of the PSGL-1 binding site for FERM (Figure 1A). 

Seven residues, including R405, T407, H408, Y410, V412, R413 and P417 on PSGL-1, were involved in ligation to the β5 strand of FERM C subdomain (Table 1, Figure 1). Of these seven residues, Y410 should be crucial, because it was contributed to the most stable H-bond (with occupancy > 95%) across the interface by pairing with I248 on FERM for all the three equilibrated structures. Paired with F250 on FERM, H408, another possible key residue on PSGL-1, was responsible for the 2nd H-bond with occupancies larger than 65% for all the three equilibrated complexes. With the occupancies of 69%, 16% and 64% for their respective equilibrated structures, the 3rd H-bond between V412 and R246 on FERM was demonstrated to be stable for the 1st and 3rd equilibrated structures and not for the 2nd one (Table 1). Additionally, contributed by R405, Y410 and P417 on PSGL-1 with their respective partners (D252, H288 and R295 on FERM), the three stable H-bonds (the 4th, 5th and the 10th bond) with occupancies larger than 60% occurred just in the 1st equilibrated conformation (Table 1). Despite the survival ratio of salt-bridge between R405 and D252 exceeded 60% only in the 1st equilibrated conformation (Table 1), R405 had been demonstrated to be crucial by mutation experiments [24]. Therefore, the equilibrated structure from run 1 here was regarded as the best stable and rational one among the three conformations in equilibrium, and was chosen as the initial conformation for the subsequent SMD simulations herein.

### 2.2. Exposure of Phosphorylation Site Y205 Was an Early Event of PSGL-1 Dissociation from FERM

To examine pull-induced unbinding of PSGL-1 from FERM, we performed the force-ramp SMD simulation over 30 ns thrice with time step of 2 fs and a pulling velocity of 5 Å/ns. The force-time curves (Figure 3A) exhibited that the tensile force on complex increased firstly with time until it reached the minor force peaks at pull time of about 12 ns for Run 1, 15 ns for Run 2 and 13 ns for Run 3, respectively, then slumped by about 25 to 35 percent, and increased again as time passed further. The breakage of the stretched PSGL-1/FERM bond occurred at the rupture force of about 250–280 pN, showing a strong resistance of the complex to stretching. This resistance to stretching would be shared mainly by the stable H-bonds, which were contributed by R405 with D252, H408 with F250, V412 with R246, Y410 with I248 and H288, and R413 with R246 and S243, respectively. The H-bond between P417 and R295 might be important for complex in equilibrium but vanished in stretching the complex with constant velocity of 5 Å/ns. At the early-middle stage of the stretching process, stretching might make the hydrogen bonds between R413 and its two partners (R246 and S243) stable (Table 1; Figure 3B), suggesting a force-enhanced H-bonding. The time courses of number of the H-bond across interface (Figure 3C) showed a triphasic dependence of H-bonding on pull time. It meant that the stretch-induced denaturation might enhance either PSGL-1 binding to or unbinding from FERM, because the H-bond numbers should be negatively correlated to the dissociation kinetics of the complex.

The nonlinear and irregular characters of the instantaneous force-time curves (Figure 3A) were closely relevant to the stretch-induced allostery or deformation of PSGL-1-ligated FERM, along with breaking and forming of the involved H-bonds. The minor force-peak in the force-time curves came from the CTL α-helix denaturation-induced dropping of tensile force. It was observed from the run 2 of the force-ramp simulations (Figure 4) that, under pulling with constant velocity of 5 Å/ns, the C-terminal linker (CTL) α-helix of PSGL-1-ligated FERM started unfolding at pulling time of 2.6 ns about and gradually became a complete unfolded one as pull time spent 12.2 ns about; and, prior to unbinding of PSGL-1 from FERM, the stretch-induced allostery of the bound FERM occurred not only in the seven-stranded β-sandwich but also in the C-terminal α1-helix (Figure 4). 

As a result, stretching could serve as an allosteric effector for decreasing the steric hindrance in phosphorylating of Y205. The solvent accessible surface area (SASA)-time curves (Figure 5) showed that the SASA value fluctuated around a low level closed to that in equilibrium state for each of ITAM-like motif and its two phosphorylation sites (Y205 and Y191) in early and middle stages with pulling time shorter than 20 ns firstly and then increased to a higher level for each of ITAM-like motif and Y205 but not for Y191 as time passed further, suggesting a stretch-induced exposing of the phosphorylation site Y205 rather than Y191 on the ITAM-like motif. Occurred at stretch force of about 200 pN, the exposure of phosphorylation site Y205 was an early event of dissociation of PSGL-1 from FERM (Figure 3A and Figure 5), suggesting that the PSGL-1/FERM complex was a stable node in the mechano-chemical signaling.

### 2.3. Tension-Induced Exposing of the Phosphorylation Site Y205 rather than Y191 in the Clamped FERM/PSGL-1 Complex Is Biphasic Force-Dependent

To further examine the regulation of tensile force on exposing of the phosphorylation sites of FERM bound with PSGL-1 in the force-clamp mode, the so called “ramp-clamp” SMD simulations were performed thrice for tensile forces of 25, 50, 75 and 100 pN (see Section 4, Materials and Methods). In each run, the complex was first pulled with velocity of 5 Å/ns until the tensile force arrived at a prescribed tensile force, and then stretched in a force-clamp mode for 100 ns. Three independent trajectories starting from three different initial conformations were generated for the complex under each of the given tensile forces, because the multiple short-trajectories might be more effective than a single long-trajectory in capturing the protein dynamics [32]. It was believed that in a force-ramp simulation, the complex was stretched too fast and driven out of equilibrium, leading to a hysteresis of conformational change [33]. This allosteric hysteresis might be reduced by performing a force-clamp simulation, in which the complex could spend an adequate time to respond to stretching and reach to an equilibrium under loading. 

We observed from the force-clamp simulations that the CTL α-helix of the bound FERM would be unfolded sufficiently within 5 ns about, just with a slight deformation of other stretched subdomain of the complex (Appendix A), but the rigid body motion of C domain relative to A and B domain of the stretched FERM was obvious (Figure 6). It was showed that, the redistribution of the structural unbalanced-strain built-up at the force-ramp stage was illustrated by increasing of the C_α_ root mean square deviation (RMSD) of the clamped complex obviously with tensile force but slowly with the relaxing time (Appendix A), and the conservation of the seven-stranded β-sandwich and the C-terminal α-helix of FERM domain C was reflected by the negligible relative extension (<0.8 Å) of the clamped subdomain of the complex (Appendix A and Figure 6; Materials and Methods section), based on the quasi-complete sample space of the stretched complex with Gaussian distribution of the number of H-bonds on interface (Appendix A).

The tension-induced allostery rather than the rigid-body motion of FERM C domain should be contributed to exposing of phosphorylation site Y205, but the less effect of tensile force on the buried phosphorylation site Y191 in FERM B domain came from the site far away from the force transduction pathway in the loading strategy (Appendix A and Materials and Methods section). Plots of the mean SASA of ITAM-like motif and its two phosphorylation sites Y205 and Y191 against tensile force showed that increasing force made the SASA of ITAM-like motif increased first and then decreased as the tensile force increased, and so did the SASA of Y205, but the turn points of the tensile force located at 50 pN for ITAM-like motif and 75 pN for the phosphorylation site Y205; and the SASA of Y191 remain almost small for various tensile forces, as it should be (Figure 6 and Figure 7). The tension-induced SASA increments might run up to 143 Å^2^ for ITAM-like motif and 26 Å^2^ for Y205, in comparison with the mean SASA values of 668 Å^2^ for ITAM-like motif and 43 Å^2^ for Y205 at zero tensile force. These data suggested a significant force-promoted exposing of the phosphorylation site Y205 under loading with tensile forces below threshold of 75 pN. 

### 2.4. Unbinding and Allostery of the FERM Bound with PSGL-1 at Various Constant Tensile Forces 

To check whether the tensile force regulates binding of PSGL-1 to FERM or not, the H-bonds (and/or salt bridges) across binding site and the interaction energies of the complex under constant tensile forces were sampled through the above mentioned “ramp-clamp” SMD simulations (Section 4, Materials and Methods). Plots of the mean interaction energy (*E*) and the mean number of the H-bonds (*N*_HB_) (and/or salt bridges) across binding site over 100 ns for the thrice runs against the constant tensile force (*F*) showed that the mean interaction energy (*E*) decreased first and then increased with the tensile force, and the force threshold occurred at 50 pN (Figure 8A), illustrating a biphasic force-dependent binding affinity of FERM to PSGL-1; and, on the contrary, the mean number of the hydrogen bonds (*N*_HB_) across binding site increased first and then decreased with the tensile force, but the force threshold remained same (Figure 8B), because the more the H-bonds across binding site, the stronger the binding of FERM to PSGL-1. As a result, increasing tensile force made *f_D_*, the normalized complex dissociation probability or the mechano-regulation factor on complex dissociation, decrease first until force reached at its threshold of 50 pN, and then increase (Figure 8C), suggesting a catch-slip bond transition in dissociation of PSGL-1 from FERM. This phenomenon of the catch-slip bond transition had been observed through AFM and BFP as well as flow chamber experiments for various adhesion molecular systems, such as P-, E-, or L-selectin with PSGL-1 [34,35,36], von Willebrand factor (VWF) with platelet glycoprotein Ibα (GPIbα) [37] or ADMAMTS13 (A Disintegrin and Metalloprotease with thrombospondin motifs-13) [38], and so on. 

The force-induced allostery was believed to be responsible for the “catch-slip bond” mechanism mentioned above for the complex of FERM and PSGL-1. We measured *θ*, the mean angle between β1C and β4C of FERM C domain over simulation time of 100 ns thrice under each constant pull force to scale the force-induced allostery of the bound FERM (Figure 9). The angle *θ* increased remarkably first and then decreased with *F*, the force turning point shared with *f_D_*, *E* and *N*_HB_ (Figure 8). The plots of the C_α_ root mean square fluctuation (RMSF) for PSGL-1(Figure 10A) peptide and FERM C domain (Figure 10B) against tensile force showed that increasing force make PSGL-1 peptide flexibility lower first and then higher, and the flexibility of the FERM C domain remained almost constant under tensile force ≤ 50 pN but increased at tensile force ≥ 75 pN, because that the RMSF pattern could mark the local structural flexibilities. These data (Figure 8, Figure 9 and Figure 10) demonstrated that, the angle *θ* was corelative negatively to the normalized complex dissociation probability *f_D_*, the interaction energy (*E*) and the flexibility of the ligated PSGL-1 peptide, but positive to the H-bond number (*N*_HB_), as it should be. It suggested that the force-induced allostery might be responsible for the biphasic force-dependent unligation and phosphorylation of ERM bound with PSGL-1.

### 2.5. The Key Residues Were Responsible for the Force-Regulated Interaction of PSGL-1/FERM Complex

To understand the structural basis of “catch-slip bond” transition mechanism for dissociation of PSGL-1 from FERM, we examined the H-bonds between residue pairs on binding site through “force-clamp” SMD simulations under tensile forces of 0, 25, 50, 75 or 100 pN, and evaluated the probabilities of either the FERM residues bound to the PSGL-1 or the PSGL-1 residues bound to the FERM through Equations (2) or (3) (see Section 4, Materials and Methods). We observed from the simulations (Figure 11, Appendix A) that, under loading forces, the sixteen detected H-bonds across complex interface were contributed by residue pairs, such as H408 with F250, V412 with R246, R405 with D252, N251 with K406 and T407, Y410 with its three partners (N247, I248 and H288), P417 with its four partners (R246, N247, R295 and K296), and R413 with its four partners (S243, I245, R246 and H288), respectively; and, of these sixteen H-bonds, the dominant ones with survival ratios larger than 50% were those between H408 and F250, Y410 and I248, R405 and D252, P417 and R295, and V412 and R246 under various tensile forces. 

Under loading, the these H-bonding events were multifariously force-dependent and might become strong or week in responding to the increasing tensile force, accompanying with forming or vanishing of some interactions; and, five residue interaction modes, so called as the “steady”, “catch-slip”, “slip-catch”, “catch-slip-catch” and the “slip-catch-slip” type, were shown in their respective typical residue interactions, such as Y410 paired with I248, P417 with K296, Y410 with N247, R413 with R246 and P417 with R295 (Figure 11). It suggested that the cooperative interaction of the H-bonds in different mechano-responding modes mediated the force-induced change of conformation and function of PSGL-1 bound with FERM. It was believed that Y410, H408 and R405 with their respective partners, such as I248 and F250 as well as N251, should be responsible for the mechano-stability of the complex, they contributed the residue interactions in top three came from, respective, for the high occupancies (>0.65, almost for each tensile force) of the involved H-bonds. Interesting, tensile force made the identified the R405 [24] crucial more.

## 3. Discussion

In the selectin-induced recruitment of flowing leukocytes to vascular injury sites, the flow-enhanced cell adhesion was believed to be mediated through the interaction of PSGL-1 with selectin [1,2]. This interaction was governed by a “catch-slip bond transition” mechanism, which said a force-induced prolongation of the bond lifetime for PSGL-1/selectin complex under fluid shear stresses below a force threshold [36]. A similar “catch-slip bond transition” phenomenon (with force turn point of 50 pN) for the interaction of PSGL-1 with FERM was simulated here by a series of “ramp-clamp” mode SMD simulation under various mechanical loads (Figure 8). It suggested a force-enhanced biophysical transmembrane connectivity or a transmembrane mechano-stability of the PSGL-1 signal pathway in flow, through a same mechanism of force-induced prolongation of the bond lifetime for complex of PSGL-1 with either of selectin and FERM. Additionally, a high mechanical strength of adhesive molecule bond between FERM and PSGL-1 should be required for resisting the external force-induced rupture of the complex and for maintaining a high-level biophysical connectivity of the PSGL-1 signal pathway. This requirement was observed from the SMD simulations in ramp mode with constant pulling velocity, because of a high rupture force (>250 pN) of the pulled complex (Figure 3). The high mechanical strengths of adhesive molecular bond between PSGL-1 and FERM might be responsible for the stable mechano-chemical signaling along the PSGL-1 pathway. Being relevant closely to the catch-slip bond transition (Figure 8), the force-induced allostery of FERM bound with PSGL-1 peptide (Figure 9) might be required for resisting to the pull-induced breakage of the adhesive bond, because of the allostery-induced drop of tension on the stretched complex (Figure 4).

Exposure of phosphorylated sites in the ITAM-like motif is crucial for PSGL-1 signaling, because the ITAM-like motif required be phosphorylated for binding to SH2 domain of Syk [39,40]. The crystal structure analysis for the complex of PSGL-1 with ERM protein at equilibrium indicated that the complex was stable, mainly coming from two strong hydrogen bonds (Y410 paired with I248 and H408 paired with F250) with high survival ratios > 90% (Table 1); this high conformational stabilization showed a strong space steric effect, which made unembedding of the phosphorylation sites on the ITAM-like motif of the ERM protein difficult (Figure 1, Figure 7 and Appendix A). However, this space steric effect should be weakened by the force-induced allostery of the FERM bound with PSGL-1 (Figure 4 and Figure 6). We found from SMD simulations that, in ramp-mode with pulling with constant velocity, increasing tensile force would make the embedding phosphorylation sites in ITAM-like motif of the PSGL-1 peptide-ligated FERM exposed (Figure 5), and in “ramp-clamp” mode at constant tensile force of 0, 25, 50, 75 or 100 pN, increasing tensile force enhanced first and then weakened exposing of ITAM-like motif and its phosphorylation site Y205 in the ligated FERM with force threshold of 75 pN (Figure 7). The tensile forces on complex might come either from the flow loading on the cells tethered or adhered to vessel wall, or from tension in the rearranged cytoskeleton [23]. It suggested that, the steric hindrance effect provides a strong resistance to the exposure of the embedding phosphorylation sites in ITAM-like motif [23], but should be weakened by mechano-chemical signal, which is received by the extracellular domain of PSGL-1 ligated to the immobile selectins on vessel for “Outside-In” signaling or through the cytoskeleton bound with ERM for “Inside-Out” signaling [3,4] and transduced to the ERM protein ligated with both the intracellular PSGL-1 peptide and the cytoskeleton. However, increasing tensile force over force threshold of 75 pN might prompt the phosphorylation site Y205 to be reburied into the ligated FERM (Figure 7), possibly coming from the force-induced allostery of the PSGL-1 peptide-ligated FERM, breakage of some H-bonds in complex interface and so on (Figure 6, Figure 7 and Appendix A). In this simulation, Y191 was almost always embedded in the complex, because the residue Y191 was located at the FERM B domain, which was far away from both the binding site of the complex and direction of the tensile force (Figure 6).

However, in MD simulation for receptor-ligand interaction in mechano-microenvironment, the timescale effects on the results at atom level led to a barrier in predicting mechano-chemical processes within time scale of sub-seconds, because that these biomolecular processes, such as the conformational evolution of protein and interaction of receptor with ligand, would undergo a period far longer than the simulation time, which range from nanoseconds to milliseconds. Besides, in comparison with the single-molecule atomic force microscopy and the optical tweezers experiments, the MD simulation usually have either a pulling speed of three to four orders of magnitude or a tensile force of one to two orders of magnitude higher than those in AFM or optical tweezers experiments [41,42]. Thus, the simulation results usually could not be compared with experiment data and repeated by other one. A rational reason may be that the simulation results were derived from an unperfect conformational sample space. With the assumption that samples in a perfect space is normal if the system is under white noise simulation, the data of the *N*_HB_ on binding site were used to test whether the conformational sample space was normal or not in this work. The *N*_HB_ distribution was observed to be quasi-normal because it was converged to 7.7 ± 1.0 as the simulation time was over 100 ns (Figure 2), suggesting that the minimum sampled time for forming a perfect conformational space was not overlong so that a less time-consuming compute strategy could be expected in MD simulation.

In summary, we here demonstrated a regulation of tensile force on both phosphorylation site exposure and unligation of FERM bound with PSGL-1 peptide through a series of SMD simulations. Increasing tensile might prompt PSGL-1 signal through enhancing both the interaction of PSGL-1 to FERM and the phosphorylation site exposure of the ligated FERM in flow. This mechano-chemical process of the complex should be mediated by the force-induced complex allostery, and the key involved residues might include Y410, H408, R405 and P417 on PSGL-1 with their respective partners, such as I248, F250 and N251 as well as R246 (Figure 11). The present work might provide a novel insight into PSGL-1 signaling in various cellular immune responses under mechano-environments. 

## 4. Materials and Methods

### 4.1. System Setup 

The crystal structure of PSGL-1/FERM complex was read from the PDB database (Protein Data Bank code 2EMT; Resolution: 2.8 Å), being composed of the intracellular domain near the membrane of PSGL-1 (residues 402–417) and the radixin protein FERM domain (residues 1–313), which contains three subdomains (A, B and C) and C-terminal linker (CTL) (Figure 1) [43,44]. The domain C of FERM consists of a seven-stranded β-sandwich and the C-terminal α-helix. The hydrophobic core of the domain C was enclosed by the seven-stranded β-sandwich and packed with the C-terminal α-helix over β5 and β1 strands. Two software packages, VMD 1.9.2 and NAMD 2.12 [43,44], were used for visualization and modeling, respectively. The complex was soaked with TIP3P [45,46] water molecules in a rectangular box (165.6 Å × 96.8 Å × 95.6 Å) with walls at least 15 Å away from any protein atom. The system was neutralized with 150 mM NaCl to mimic the actual physiological environment [47], being consisted of 145,099 atoms (Appendix A). 

### 4.2. Molecular Dynamics Simulations

The CHARMM27 all-atom force field [48,49], along with cMAP correction for backbone, particle mesh Ewald (PME) algorithm [50] for electrostatic interaction, a 12 Å cut off for electrostatic and van der Waals interaction, was used here to perform MD simulations with periodical boundary condition and time step of 2 fs. The system was minimized firstly at 5000 steps with heavy or non-hydrogen protein atoms being fixed, and then minimized at 5000 steps with all atoms free. The energy-minimized systems were heated gradually from 0 to 310 K in 0.1 ns first and then equilibrated thrice for 100 ns with pressure and temperature control. The temperature was held at 310 K using Langevin dynamics, and the pressure was held at 1 atmosphere by the Langevin piston method. The best stable one of the three complex structures in equilibrium was chosen as the initial conformation for the subsequent steered molecular dynamics (SMD) simulations [51]. 

The so-called “ramp-clamp” SMD simulations, a force-clamp MD simulation followed a force-ramp one, were performed on the equilibrated system to examine the force-induced unbinding and conformation changing of the FERM bound with PSGL-1. The C-terminal C_α_ atom of FERM (S313) was fixed, and the N-terminal C_α_ atom of PSGL-1 (R402) was steered along pulling direction perpendicular to the binding surface of the complex. The virtual spring, connecting the dummy atom and the steered atom, had a spring constant of 69.48 pN/nm. The complex was pulled over 30 ns thrice with time step of 2 fs and a constant velocity of 5 Å/ns, at which the pulling would be contributed to hydrogen bond (H-bond) rupture with conservation of secondary structures of the complex [52] (Appendix A). Once tensile force *f* arrived at a given value, such as 25, 50, 75 or 100 pN, the SMD simulation was transformed from the force-ramp mode to a force-clamp one, at which the complex was stretched with the given constant tensile force for the followed 100 ns (Appendix A). Each events of hydrogen bonding under stretching were recorded to examine the involved residues and their functions. 

### 4.3. Data Analysis

All analyses were performed with VMD tools [53,54]. We measured the C_α_ root mean square deviation (RMSD) and the solvent accessible surface area (SASA) (with a 1.4 Å probe radius) to characterize the conformational change and the hydrophobic core exposure, respectively. To mimic, in another way, the change of the complex structure under stretching, we here defined Δ*L*, a relative extension of the stretched subdomain of the complex, by:(1)$$\Delta L=\left(L-L_{0}\right)/L_{0}$$
where, *L*_0_ and *L* were the distances between the steered-C_α_ atom on the PSGL-1 N-terminal residue R402 and the fixed C_α_-atom on the C-terminal residue S313 of FERM C domain at force-clamp mode with zero and nonzero constant tensile force, respectively. 

A hydrogen bonding event occurred once the donor-acceptor distance and the donor- hydrogen-acceptor angle were less than 3.5 Å and 30°, respectively. A salt bridge was defined if the distance between any of the oxygen atoms of acidic residues (Asp or Glu) and the nitrogen atoms of basic residues (Lys or Arg) must be within 4 Å. An occupancy (or survival ratio) of a hydrogen bond (H-bond) or a salt bridge was evaluated by the percentage of bond survival time in simulation period. As a reflection of the receptor-ligand binding affinity [55], the rupture force was read from the maximum of the force spectrum in a force-ramp MD simulation with constant pulling velocity. All visual inspections and molecular images were completed by using VMD 1.9.2. 

To estimate the residue-residue interactions across binding site through hydrogen bonding, we herein introduced *p_ij_*, the probability of the *i*th ligand residue binding with the *j*th receptor residue, which was defined by:(2)$$p_{ij}=1-\prod_{l=1}^{M_{ij}}\left(1-\omega_{ij,l}\right),\;\;i=1,2,\ldots ,M_{L};j=1,2,\ldots ,M_{R};\;j=0,1,\ldots ,M_{ij}$$
where, the formation or breakage of each H-bond on binding site was assumed to be an independent event not related to other bonds, *ω_ij,l_* is the survival ratio of the *l*th H-bond between the *i*th ligand residue and the *j*th receptor residue. *M_R_* (≥1) and *M_R_* (≥1) are respectively the total numbers of ligand and receptor residues involved in binding, and *M_ij_* (≥0) expresses the numbers of H-bonds between the *i*th ligand residue and the *j*th receptor residue. *P_j,L_* (the probabilities of the *j*th ligand residue binding to the receptor) and *P_j,R_* (the probabilities of the *j*th receptor residue binding to the ligand) are respectively calculated approximately by:(3)$$P_{j,L}=1-\prod_{i=1}^{M_{R}}\left(1-p_{ji}\right)\;\mathrm{and}\;P_{j,R}=1-\prod_{i=1}^{M_{L}}\left(1-p_{ij}\right)$$

Furthermore, *P_D_*, the dissociation of ligand from receptor, could be estimated by:(4)$$P_{D}=\overset{M_{L}}{\underset{j=1}{\prod }}\left(1-P_{j,L}\right)=\overset{M_{R}}{\underset{j=1}{\prod }}\left(1-P_{j,R}\right)$$

It was a great challenge to overcome the barrier of the effect of timescale difference of 5–7 order of magnitudes led. We herein introduced *f_D_*, the mechano-regulation factor, which was ratio of *P_D_* at tensile force of *f*_0_ and of *P_D_* at zero tensile force, that is:(5)$$f_{D}=P_{D}{|}_{f=f_{0}}/P_{D}{|}_{f=0}$$
where *f* expressed the tensile force on the complex. However, there might be a significant gap between the results from MD simulation and the data measured with single-molecular tools, such as atomic force microscopy (AFM), optical and magnetic tweezers [56], coming from effects of timescale on predicting ligand-receptor interaction with a timescale of about 0.01–1.00 s by MD simulation of about 100 ns. Regardless the timescale effect on complex dissociation *P_D_*, it was expected here that *f_D_*, the mechano-regulation factor or the normalized complex dissociation probability, should be comparable with experiment data if the conformations sampled from simulation is perfect.

### 4.4. Statistical Analysis

Differences between groups were disclosed by using Ordinary one-way ANOVA nonparametric analysis of variance followed by Tukey’s multiple comparisons test. *p* values of < 0.05 were considered significant.

The ensemble workflow of computational procedure was shown in Appendix A.

## Figures and Tables

**Figure 1 ijms-21-07064-f001:**
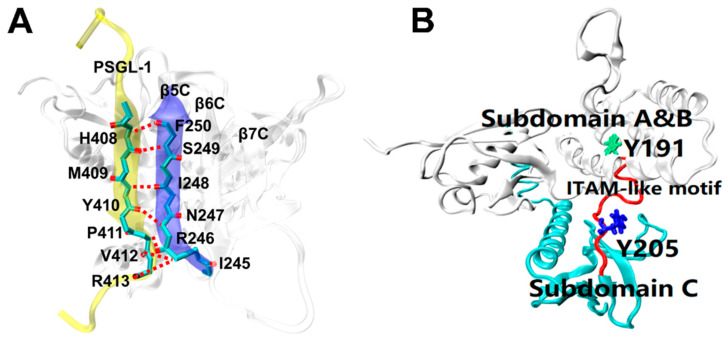
Crystal structures of FERM domain bound with or without PSGL-1 peptide [24]. (**A**) Main chain–main chain interaction between the PSGL-1 peptide (yellow, NewCartoon) and strand β5C (blue, NewCartoon) of FERM domain (white, NewCartoon) (PDB ID: 2EMT). Involving this interaction, the hydrogen bonds are shown as red broken lines, the linkers of the residues, such as H408, M409, Y410, P411, V412 and R413) on PSGL-1 peptide to their partners on the β5C of FERM C domain. (**B**) The ITAM-like motif (red, Tube) and its two phosphotyrosine-binding sites Y191 (green, Licorice) and Y205 (blue, Licorice) in FERM domain. Y191 and Y205 are buried in subdomain B and C of FERM, respectively.

**Figure 2 ijms-21-07064-f002:**
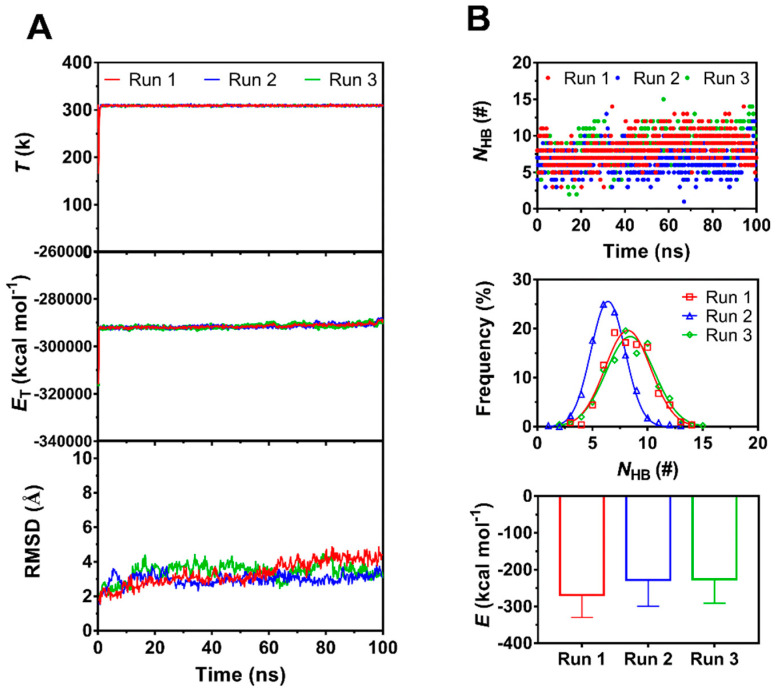
The time courses of temperature (*T*), total energy (*E*_T_)in system and number of H-bonds (*N*_HB_) between the interfaces and root mean square deviation (RMSD) of heavy atoms of the complex, the mean interaction energies (*E*) of FERM with PSGL-1 over 100 ns, and the distributions of H-bonding events within 100 ns at binding site for thrice 100 ns equilibriums. (**A**) The time courses of *T*, *E*_T_ and RMSD. Both *T* and *E*_T_ remained almost constant for each run of 100 ns, and RMSD was limit in region from 2 Å to 4 Å, showing a equilibrate system for each run. (**B**) The interaction energies (*E*), the time courses and frequencies of *N*_HB_ for three runs. The best fitting data showed a Gaussian distribution of *N*_HB_ frequency for each run, and, the maximum *N*_HB_ and the minimum *E* were obtained in Run1, suggesting the equilibrated complex form Run1 was the best one for subsequent simulation.

**Figure 3 ijms-21-07064-f003:**
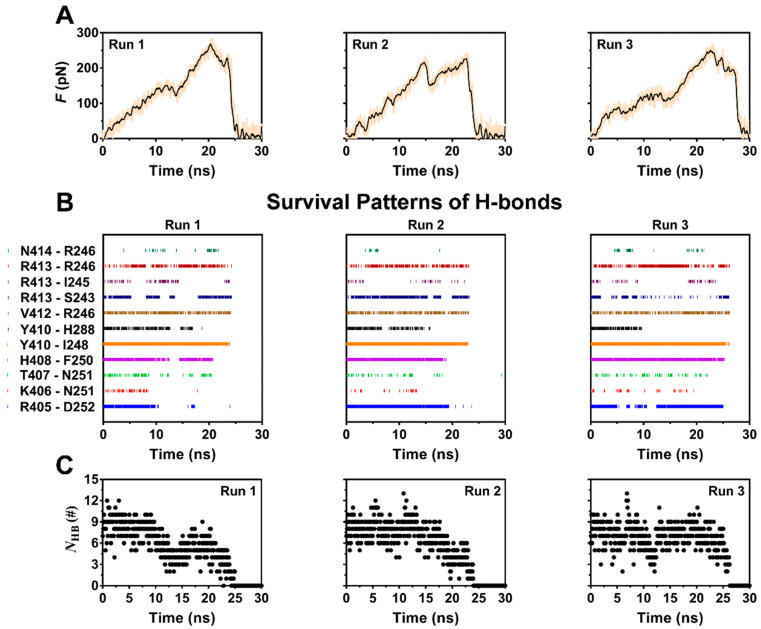
Time-courses of pull force, the survival pattern and number of H-bonds along the dissociation pathway of the complex for three runs with constant velocity of 5 Å/ns. (**A**) The time curves of loading force on complex for three runs. The rupture forces were over 200 pN for each run, showing a high mechanical strength of the complex. (**B**) The survival patterns of H-bonds across complex interface for three runs. In each pattern, the colored and uncolored line expressed the unbroken and broken state of the bond, respectively. These different survival pattens of the eleven involve H-bonding events also showed the diverse dissociation pathways of complex under pulling. (**C**) The number of the H-bond (*N*_HB_) across interface for three runs. The diverse stretch-induced dissociation of ERM from PSGL-1 was suggested by the *N*_HB_ patterns, in which, the valley points were matched well to the relevant peak points of the relevant force patterns (Figure 3A).

**Figure 4 ijms-21-07064-f004:**
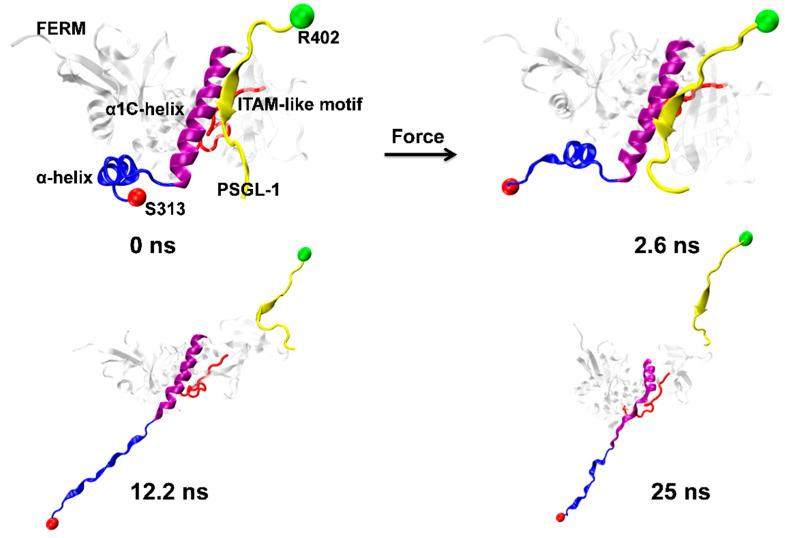
Pull-induced allostery of the PSGL-1/FERM complex (Snapshots) under pulling. The typical snapshots of four complex intermediates from Run 2 at four different simulation times of 0.0, 2.6, 12.2 and 25 ns. Pull-induced Unfolding of the α-helix (blue, NewRibbons) occurred at time of 2.6 ns, but the α1C-helix (purple, NewRibbons) were unfold at time of 12.2 ns. Partially unburying of ITAM-like motif (red, Tube) were observed at simulation time of 12.2 ns and 25 ns, at which PSGL-1 was dissociated from FERM.

**Figure 5 ijms-21-07064-f005:**
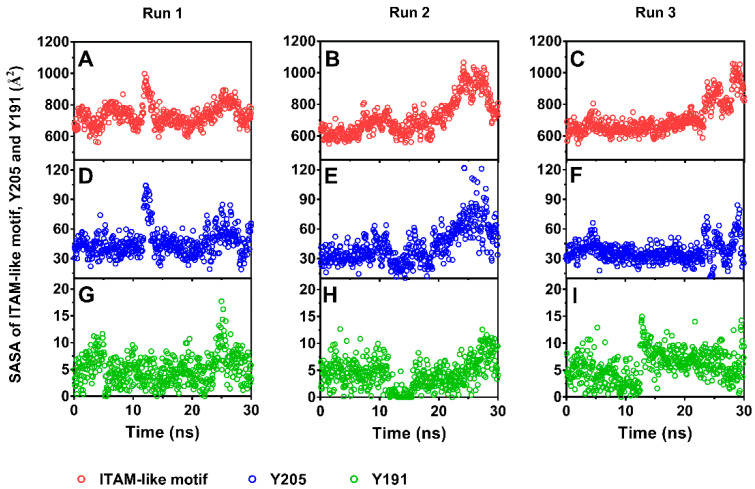
Time courses of solvent accessible surface areas (SASAs) of immunoreceptor tyrosine activation motif (ITAM)-like motif (red dot) and phosphorylation site Y205 (blue dot) and Y192 (green dot) under pulling with constant pull velocity. SASA values of the ITAM-like motif (**A**–**C**) and Y205 (**D**–**F**) increased significantly as time passed through times of about 25 ns, but SASA values of Y191 (**G**–**I**) remain a very low level in comparison with Y205.

**Figure 6 ijms-21-07064-f006:**
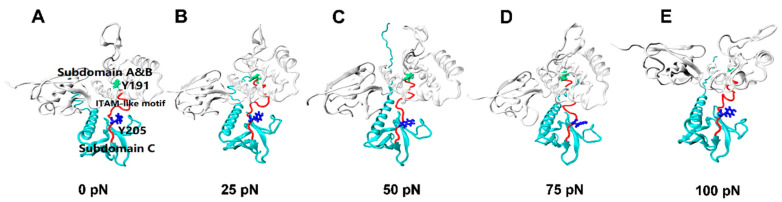
Conformational changes of FERM domain (white, NewCartoon) during constant force SMD at a level of 0 pN(**A**), 25 pN(**B**), 50 pN(**C**), 75 pN(**D**), and 100 pN(**E**). These structural changes can be visually described by a dynamic evolutionary process, with each conformation taken from a 100 ns constant force tensile simulation of the PSGL-1/FERM complex. Exposure of ITAM-like motif (red, Tube) and phosphorylation site Y205 (blue, Licorice) were presented in the graph, phosphorylation site Y191 (green, Licorice) was buried all the time.

**Figure 7 ijms-21-07064-f007:**
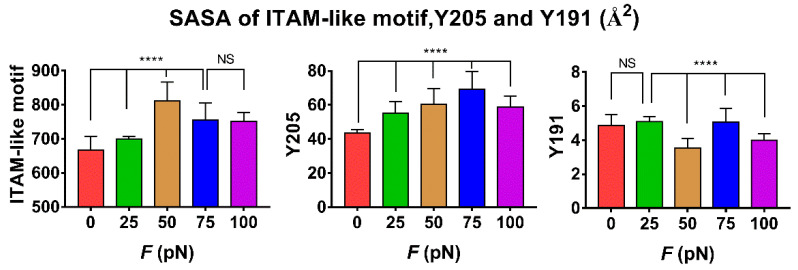
Variation of SASA of ITAM-like motif, phosphorylation site Y205 and Y191 versus tensile force *F*. All data were averaged over simulation time of 100 ns for three runs. Differences between groups were disclosed by using Ordinary one-way ANOVA nonparametric analysis of variance followed by Tukey’s multiple comparisons test. Results represent the mean ± S.E. (error bars) of three independent simulations. (****, *p* < 0.0001, NS, No statistical difference).

**Figure 8 ijms-21-07064-f008:**
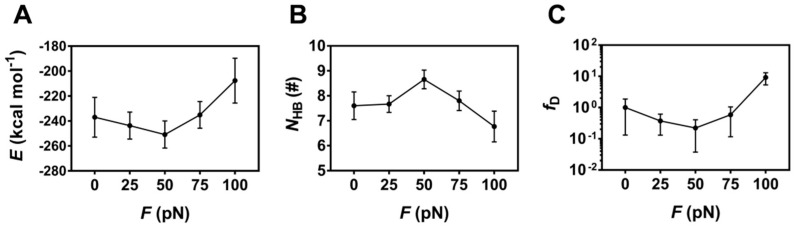
Variation of the mean binding energy *E*, the mean hydrogen bond number *N*_HB_ on the interface and the mean relative dissociation probability *f_D_* versus tensile force *F*. Plots of *E* (**A**), *N*_HB_ (**B**) and *f_D_* (**C**) over 100 ns for three runs against tensile force showed a biphasic regulation of tensile force on *E*, *N*_HB_ and *f_D_*. All data were shown with means ± S.E. (*n* = 3).

**Figure 9 ijms-21-07064-f009:**
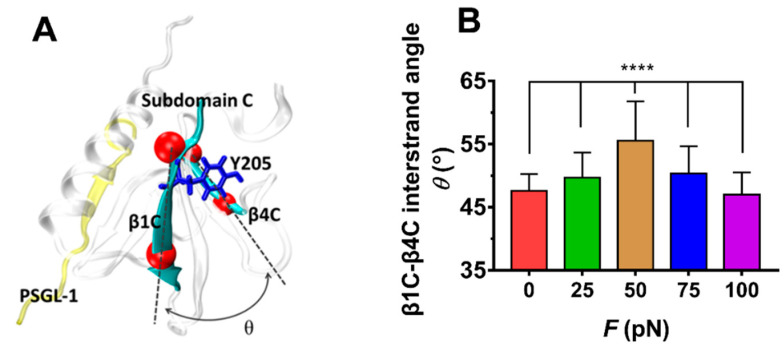
Variation of the mean β1C-β4C interstrand angle *θ* of the FERM C Subdomain bound with PSGL-1 peptide versus tensile forces. (**A**) The FERM C Subdomain (grey) bound with PSGL-1 peptide (yellow). In structure, the angle *θ* between β1C and β4C sheet (cyan) were defined by the cross angle of two straight lines between the C_α_-atoms (red) at the two ends of sheet β1C and β4C. The conformation was averaged over 3 × 100 ns involved in three runs with force-clamp simulation time of 100 ns. (**B**). Plot of *θ* against tensile force *F*. the mean β1C-β4C interstrand angle *θ* was averaged over the entire simulation duration of 100 ns for each the three runs. The *p*-values of the Tukey’s multiple comparisons test were shown to indicate the statistical difference significance (**** *p* < 0.0001), or lack thereof. All data shown are means ± S.D., *n* = 3.

**Figure 10 ijms-21-07064-f010:**
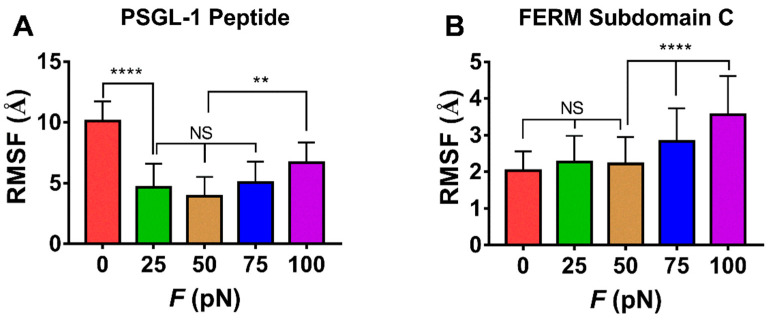
The force-mediated flexibilities of PSGL-peptide (residues 402–417) and FERM C subdomain (residues 199–297). Variation of the mean RMSF over the entire force-clamp simulation duration of 100 ns for each the three runs for PSGL-1 peptide (**A**) and the FERM C domain (**B**) against tensile force. All data shown are means ± S.D., *n* = 3. The *p*-values of the Tukey’s multiple comparisons test were shown to indicate the statistical difference significance (** *p* < 0.01, **** *p* < 0.0001), or lack thereof. All data shown are means ± S.D., *n* = 3.

**Figure 11 ijms-21-07064-f011:**
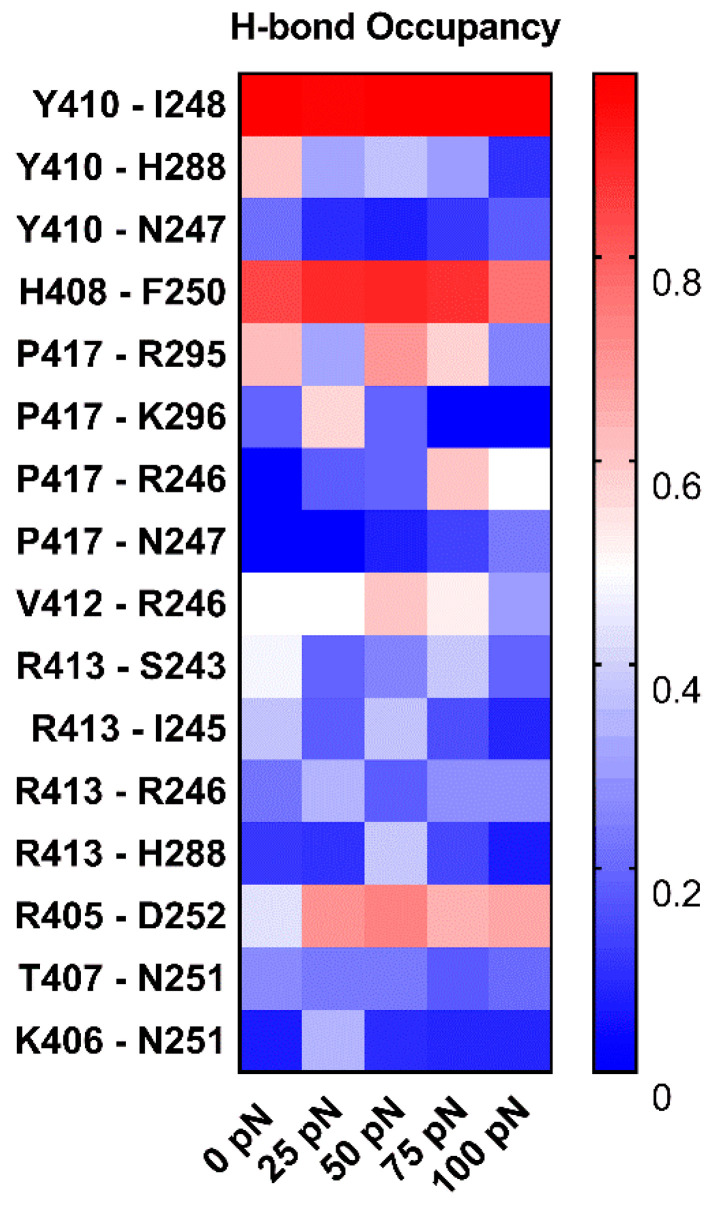
Variation of the mean occupancies of sixteen detected H-bonds over 3 × 100 ns (the entire force-clamp simulation for three runs with duration of 100 ns) against tensile force. The color bar marked the values of the H-bond occupancies. The H-bond occupancy patterns showed the various force-dependent residue interactions in complex interface. The residue interactions in top three came from Y410, H408 and R405 with their partners, I248, F250 and N251, respective, for the high occupancies (>0.65, almost for each tensile force) of the involved H-bonds.

**Table 1 ijms-21-07064-t001:** The top ten H-bonds across complex interface and their occupancies in equilibrium.

Ranking	PSGL-1 Residues	FERM Residues	Mean Occupancy			Average
			Run 1	Run 2	Run 3	
1	Y410	I248	0.99	0.97	0.98	0.98 ± 0.01
2	H408	F250	0.67	0.90	0.94	0.84 ± 0.15
3	V412	R246	0.69	0.16	0.64	0.50 ± 0.29
4	R405	D252	0.80	0.25	0.23	0.43 ± 0.32
5	Y410	H288	0.66	0.00	0.53	0.40 ± 0.35
6	R413	S243	0.43	0.04	0.50	0.32 ± 0.25
7	T407	N251	0.16	0.23	0.41	0.27 ± 0.13
8	R413	I245	0.37	0.01	0.38	0.25 ± 0.21
9	R413	R246	0.23	0.25	0.19	0.22 ± 0.03
10	P417	R295	0.62	0.00	0.00	0.21 ± 0.36

The H-bond occupancies in column 4–6 showed a mean over simulation time of 100 ns, and these data from three runs were further averaged and shown as means ± S.D. (see the column 7). The fourth bond between R405 and D252 was a salt bridge.

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
