# Peer review of "Biphasic Force-Regulated Phosphorylation Site Exposure and Unligation of ERM Bound with PSGL-1: A Novel Insight into PSGL-1 Signaling via Steered Molecular Dynamics Simulations"

_ijms, 2020, doi:10.3390/ijms21197064_

Round 1

Reviewer 1 Report

The manuscript is focused on the study of transmembrane protein PSGL-1 and its signaling. 

The paper is using MD to study interaction dynamics and post-translational modification. The paper is well written although some figures are having low quality such as figure 1 & 4. Please improve that

The authors also compared the MD results to other methods such as AFM and optical tweezers. Since the authors are interested in describing dynamics on the different time-scale, I strongly recommend adding a section about how using  CHARMM27 force field could be compared to NMR techniques which can cover a wide range of dynamics and if there are any NMR studies done on this system. 

Author Response

comment 1: “……. The paper is well written although some figures are having low quality such as figure 1 & 4. Please improve that”

Answer: Thanks for this suggestion. We havd replotted Figure 1 and 4 in the Revised version in which the quality of these replotted Figures has higer quallity than those in orignal version.

Comment 2: “....... Since the authors are interested in describing dynamics on the different time-scale, I strongly recommend adding a section about how using CHARMM27 force field could be compared to NMR techniques which can cover a wide range of dynamics and if there are any NMR studies done on this system.”

Answer: Great suggestion! But we have not accepted this suggestion in the revised manuscript. this is because that, it is still challenge to gain the crystal and its structural data of stretched protein(-protein complex) through NMR techniques; and thus, the lack of structural data of stretched protein make us difficut in comparing usage of CHARMM27 force field with NMR studes. 

Reviewer 2 Report

The article presented by Jingjing Feng and coll, titled "Biphasic force-regulated phosphorylation site 2 exposure and unligation of ERM bound with PSGL-1: 3 a novel insight into PSGL-1 signaling via steered 4 molecular dynamics simulations" show the connectivity between PSGL protein and its reaction when different forces are applied. All work was done computationally and is the interpretation and analysis of three independent simulations of 100ns. In my opinion, the manuscript is too long and difficult to read by its style, but without grammatical or orthographic errors. Some figures must go to supplementary information because they aren't essential to understand the main idea of this work. I suggest reducing to 50% the number of figures (max 7 figures), and a re-writing of the results section to clarity.

Author Response

Comments: “ …… In my opinion, the manuscript is too long and difficult to read by its style, but without grammatical or orthographic errors. Some figures must go to supplementary information because they aren't essential to understand the main idea of this work. I suggest reducing to 50% the number of figures (max 7 figures), and a re-writing of the results section to clarity..”

Answer:
We have accepted reviewer 2’s suggestion partly in revising the original manuscript. In the revised manuscript,

1) We have combined Figure 3 and 5 in original manuscript to Figure 3 in the revised manuscript in which, Figure 3 A and B in revised version are same as those in the original version, but Figure 5 in original version become the Figure 3C in revised version. Thus, the sentence “The time courses of number of the H-bond across interface (Figure 5) showed a triphasic dependence of H-bonding on pull time. It meant that the stretch-induced denaturation might enhance either PSGL-1 binding to or unbinding from FERM, because the H-bond numbers should be negatively correlated to the dissociation kinetics of the complex” in line 182-186, page 6 in original version, was moved to line 165, page 5 in original version.

B) We have put Figure 8 in original manuscript into suppl Materials and named it as Figure S2. The sentences “In consistence with this observation, ……, suggesting that sampling of 100 ns is suitable in this study.” in line 237-247 on page 8 in original manuscript were simplified as “The tension-induced allostery … in the loading strategy (Figure S3; Materials and Method)” and put in front of the sentence “Plots of …” in line 243 on page 8 in revised manuscript.

C) We have moved the Figure 9 in original manuscript to suppl Materials and named it as Figure S3. The sentences “he tension-induced allostery rather than …… far away from the force transduction pathway in the Loading strategy (Figure S3; Materials and Method)” in line 254-272 on page 9 in original manuscript were simplified as “It was showed that, …, with Gaussian distribution of the number of H-bonds on interface (Figure S2-C)” in line 232-239 on page 8 in revised manuscript.

Round 2

Reviewer 2 Report

The authors complied with the request.